# Who will benefit from bariatric surgery for diabetes? A protocol for an observational cohort study

Julia S Kenkre ,[1] Ahmed R Ahmed,[2] Sanjay Purkayastha,[2] Khalefah Malallah,[2,3] Stephen Bloom,[1,4] Alexandra I Blakemore,[1,5] A Toby Prevost,[6] Tricia Tan [1]

[1]Department of Metabolism, Digestion and Reproduction, Imperial College London, London, UK
[2]Department of Surgery and Cancer, Imperial College London, London, UK
[3]Jaber Al-Ahmed Armed Forces Hospital of Kuwait, Kuwait City, Kuwait
[4]Director of Research and Development, North West London Pathology, London, UK
[5]Department of Life Sciences, Brunel University, College of Health, Medicine and Life Sciences, Uxbridge, UK
[6]Nightingale-Saunders Clinical Trials and Epidemiology Unit, King's Clinical Trials Unit, King's College London, London, UK

**Correspondence to**
Professor Tricia Tan;
t.tan@imperial.ac.uk

## ABSTRACT

**Introduction** Type 2 diabetes mellitus (T2DM) and obesity are pandemic diseases that lead to a great deal of morbidity and mortality. The most effective treatment for obesity and T2DM is bariatric or metabolic surgery; it can lead to long-term diabetes remission with 4 in 10 of those undergoing surgery having normal blood glucose on no medication 1 year postoperatively. However, surgery carries risks and, additionally, due to resource limitations, there is a restricted number of patients who can access this treatment. Moreover, not all those who undertake surgery respond equally well metabolically. The objective of the current research is to prospectively investigate predictors of T2DM response following metabolic surgery, including those directly involved in its aetiopathogenesis such as fat distribution and genetic variants. This will inform development of a clinically applicable model to help prioritise this therapy to those predicted to have remission.

**Methods and analysis** A prospective multicentre observational cohort study of adult patients with T2DM and obesity undergoing Roux-en-Y gastric bypass surgery. Patients will be comprehensively assessed before surgery to determine their clinical, metabolic, psychological, genetic and fat distribution profiles. A multivariate logistic regression model will be used to assess the value of the factors derived from the preoperative assessment in terms of prediction of diabetes remission.

**Ethics and dissemination** Formal ethics review was undertaken with a favourable opinion (UK HRA RES reference number 18/LO/0931). The dissemination plan is to present the results at conferences, in peer-reviewed journals as well as to lay media and to patient organisations.

**Trial registration details** ClinicalTrials.gov, Identifier: NCT03842475.

## Strengths and limitations of this study

► One of the first studies to prospectively and comprehensively profile patients with type 2 diabetes mellitus(T2DM) undergoing bariatric surgery to assess predictors of diabetes remission.
► Collects prospective data on factors known to be involved in the aetiopathogenesis of T2DM such as genetics and fat distribution.
► A biorepository of tissue samples will be collected to allow multi-omics analysis.
► Long-term follow-up for up to 10 years will enable assessment of the durability of remission.
► The primary limitation is that this is a preliminary study; it will identify and internally validate predictors and will require further external validation in other cohorts.

## INTRODUCTION

Currently the majority of adults globally are obese or overweight. The increasing number of those with obesity is closely mirrored by the increasing prevalence of diabetes; at present over 400 million adults are estimated to be affected, 90% of whom have type 2 diabetes mellitus (T2DM).[1] The close relationship between obesity, particularly 'central' or abdominal obesity, and T2DM has now been shown in multiple studies.[2 3] Obesity, as defined by ethnicity-specific body mass index (BMI) cutoffs, is the principal modifiable risk factor for developing T2DM. However, the weight threshold at which T2DM develops varies between individuals, with significant variation among different ethnicities exemplifying two of the important factors that are thought to influence risk: the individual pattern of where fat is deposited in the body and underlying genetic predisposition.[4–9]

Epidemiological studies have shown those with a higher waist:hip ratio (WHR), representing centrally-deposited fat, are more likely to develop the disease.[3] Central fat or higher intra-abdominal visceral fat deposition is known to be metabolically more damaging with increased likelihood of developing abnormalities in glycaemia, blood pressure and lipid profiles.[10] However, it is not simply central adiposity but also ectopic fat that may have pathophysiological consequences.[11] Evidence is accumulating regarding the detrimental effect on glycaemia from fat ectopically deposited within liver, skeletal muscle and pancreas.[12]

T2DM shows heritability and many pathogenic variants have been discovered which

influence β cell functioning and insulin resistance.[13 14] Moreover, genetic susceptibility to T2DM is known to be correlated with phenotypic changes in fat distribution: genetic variants associated with insulin resistance have been found to be correlated with lower subcutaneous adipose tissue (SAT) and a genetic predisposition to diabetes has been found in those with impaired SAT adipogenesis and increased waist circumference.[15] Combining GWAS (genome-wide association study) data to derive personalised genetic risk, while challenging, has been successfully achieved in other conditions with the potential to add further precision to a predictive score.[16 17]

## Remission of diabetes

The most effective treatment for obesity and diabetes is bariatric surgery; this has been relabelled as 'metabolic surgery' to highlight its primary utility as a therapy for diabetes and other related metabolic dysfunctions, including dyslipidaemia and non-alcoholic fatty liver disease, as opposed to primarily treating excess weight.[18] Randomised control trials have shown metabolic surgery consistently outperforms best medical therapy in improving glycaemic control and leads to durable remission in a significant proportion.[19 20] Roux-en-Y gastric bypass (RYGB), the gold-standard procedure for those with diabetes, involves the construction and anastomosis of a small pouch of stomach to jejunum such that food is diverted to the jejunum, bypassing the majority of the stomach and the duodenum. Following RYGB, remission of diabetes (as defined by markers of normoglycaemia in the absence of anti-diabetic medication) occurs in 4 in 10 patients at 1 year and this remission is long-lasting: the majority of these patients remain in remission at 5 years.[18–20] Furthermore, surgery also leads to a reduction in microvascular and macrovascular complications associated with diabetes.[21]

No single unifying mechanism found thus far can explain the improvement in glycaemic control seen in those with T2DM following RYGB. While it is clear that the 25% to 30% weight loss seen by 12 to 18 months postoperatively plays a significant role, improvement in glycaemia occurs within days following surgery, prior to any significant weight reduction, suggesting a role for weight-loss independent mechanisms.[22] Reduced calorie intake post-surgery is one such putative weight-loss independent mechanism for the immediate improvement in glycaemia as those on a very low-calorie diet can achieve reversal of the metabolic features of diabetes.[23] However, caloric restriction normally also leads to hom-eostatic compensation or counter-regulation including increased appetite, reduced energy expenditure and secretion of orexigenic gut hormones. In contrast, following surgery, appetite is known to reduce and weight-adjusted resting energy expenditure may increase.[24–26]

It is likely that caloric restriction and weight loss are only two of the many mechanisms that contribute to the postoperative metabolic improvement seen. Insulinotropic and anorexigenic gut hormones including glucagon-like peptide 1 (GLP-1), peptide YY (PYY) and oxyntomodulin increase following surgery.[27 28] PYY and GLP-1 have been implicated in the restoration of insulin secretion after RYGB.[29 30] Furthermore, infusion of such hormones leads to improved metabolic control.[31] Changes in their secretion have been correlated with postoperative metabolic and weight loss response.[30 32 33] Associated with these findings are the increased levels of plasma bile acids after surgery which have been correlated with increased levels of GLP-1.[34] Bile acids have been implicated in the improvement in glycaemia via signalling through the nuclear farnesoid X receptor and G-protein-coupled GPBAR1/TGR5 receptor.[35 36] GPBAR1/TGR5 activation leads to increased secretion of insulinotropic GLP-1 and modulates insulin sensitivity in tissues such as muscle and brown adipose tissue.[37] While changes in bile acids play a role in metabolic improvement, this still remains an area of active research.

Further adding to the complexity of the mechanism of remission, differences in the gut microbiota, that is, the microorganisms commensal within the gut, have been seen between lean and obese individuals.[38] There is some evidence that, as a regulator of the host's metabolism, the gut microbiota may be associated with obesity through several mechanisms such as enhancement of fat storage.[39] Significant changes occur in the gut microbiota following surgery[40] but the causal relationship between the microbiotal changes and metabolic improvements, that is, whether the metabolic improvements come from the microbiotal changes or vice versa, has yet to be established. Furthermore, there is a bi-directional relationship between the microbiome and bile acids: while the gut microbiota have a role in converting primary to secondary bile acids, bile acids can affect the diversity of gut microbiota.[41]

We conclude that to further explore the mechanisms of remission after metabolic surgery a comprehensive multi-omics approach must therefore be employed.

## Predicting diabetes remission following surgery

There is a substantial disparity between the numbers eligible for metabolic surgery and the resources available to deliver it; in the UK 2 million people fulfil the criteria but surgery is only being performed in 5600 patients per year.[42] Furthermore, there is variation in the metabolic response between individuals such that less than half are expected to have complete diabetes remission at 1 year following surgery. If we can better predict who will benefit from surgery by remission of diabetes we can better inform patients and direct the surgery to where it will be most effective.[43] Currently in the UK, BMI cut-offs dictate eligibility for surgery, along with more recently introduced diabetes duration and ethnicity criteria.[44] However, baseline BMI does not correlate well with the likelihood of diabetes remission thus other predictors are necessary to better inform eligibility based on intended benefit.[45]

Predictive factors have been found in multiple, mainly retrospective studies and some have been incorporated into models of diabetes remission.[46–48] These models have tried to predict diabetes remission based on easily available clinical data making them translatable to everyday practice. However, they remain limited in their clinical applicability.[49] Moreover, none have incorporated fat distribution or genetic risk factors which have important aetiopathological roles in T2DM. Furthermore, they are, in most cases, based on retrospective studies. One problem faced in this instance is that the consensus definition of diabetes remission depends on withdrawal of pharmacological therapies for T2DM for 1 year. There is often no standardised pathway to assess the remission status after surgery, which can lead to differing practice among clinicians thus hampering the ability to define remission status with accuracy.

This study will use genetic profiling, identifying genetic variants associated with fat distribution patterns, β-cell dysfunction, T2DM or insulin resistance, combined into an individual genetic-risk score for each patient.[50 51] Moreover, it will assess the distribution pattern of total body fat and markers of ectopic fat deposition within the liver, muscle and pancreas. These variables will be tested to assess if they are predictive of diabetes remission. In addition, patients will be profiled for variation in biomarkers postulated to be mechanistically involved in diabetes remission including gut hormones, microbiotal changes and bile acids.

The ultimate aim of the study will be to identify novel covariates predictive of diabetes remission after metabolic surgery, with the intention that these can be further investigated for their predictive power in larger studies and other populations.

## OBJECTIVES
### Primary objective
To develop and internally validate a predictive model of diabetes remission following metabolic surgery.

### Secondary objectives
To assess changes in diabetes and obesity-related comorbidities following surgery and investigate the mechanisms of diabetes remission.

## METHODS AND ANALYSIS
This is a prospective, multicentre observational cohort study investigating predictors of diabetes remission in patients with diabetes/pre-diabetes undergoing RYGB surgery with standardised biliopancreatic and alimentary limb lengths of 50 cm and 100 cm, respectively.

### Participants and selection criteria
Study participants will be recruited according to the inclusion and exclusion criteria detailed in table 1.

**Table 1** Inclusion and exclusion criteria

| | |
|---|---|
| Inclusion criteria | Patients with pre-diabetes or diabetes eligible for bariatric/metabolic surgery on the NHS, that is, BMI ≥35 with a comorbidity or ≥40, lower BMIs accepted if of Asian origin or recent-onset diabetes, referred to Imperial Weight Centre or Chelsea and Westminster NHS Trust[44 55]<br><br>Stable weight (≤10% variation) for at least 3 months |
| Exclusion criteria | Current pregnancy<br><br>Inability to give informed consent<br><br>Type 1 diabetes, low fasting C-peptide, secondary diabetes or absence of β-cell function<br><br>Unable to undergo DEXA<br><br>Cirrhosis, ascites or other condition that may modify body fat composition, for example, underlying malignancy<br><br>Participation in a concurrent/recent interventional trial within the last 3 months that would affect the current study results or mean that undergoing the study would be too burdensome for the participant<br><br>Unable to understand English |

BMI, body mass index; DEXA, dual-energy X-ray absorptiometry; NHS, National Health Service.

## Outcomes
The primary outcome is diabetes remission (partial and complete) or diabetes remission (complete) measured postoperatively at 15 months. This will be defined by American Diabetes Association criteria, such that complete remission is fasting glucose <5.6 mmol/L, glycated haemoglobin (HbA1c) <42 mmol/mol on no anti-diabetic agents, and partial remission is defined as fasting glucose 5.6–6.9 mmol/L, HbA1c <48 mmol/mol on no anti-diabetic agents. Patients will be seen at 3 months as part of routine clinical follow-up and will have anti-diabetic medication stopped according to a standardised pathway. An assessment of diabetes remission will also be made at 1, 2, 3 and 5 years.

In addition to the primary outcome we will collect phenotyping data to allow assessment of diabetes severity and obesity associated comorbidities.

## Exploratory outcomes
► Change in epigenetic data including DNA methylation, histone modifications and non-coding RNA in blood, liver, muscle, fat (subcutaneous and visceral) and stomach and small bowel and spermatozoa.
► Changes in the transcriptome in blood, liver, muscle, fat (subcutaneous and visceral) and stomach and small bowel and spermatozoa.
► Changes in gut hormone profiles in response to a mixed meal.

**Table 2** Time and events table of measures that participants will undergo throughout the study

| Visit number | 1 | Surgery | 2 | 3 | 4 | 5 |
|---|---|---|---|---|---|---|
| Month | Preoperative covariates measured here | 0 | 15 Primary outcome measured here | 24 | 36 | 60 |
| Clinical assessment. Anthropometrics* and bioimpedance measurement | ✓ | Weight only | ✓ | ✓ | ✓ | ✓ |
| Blood tests including HbA1c and fasting glucose to enable assessment of remission† | ✓ | ✓ | ✓ | ✓ | ✓ | ✓ |
| Mixed meal test | ✓ | | ✓ | | | |
| Psychological and exercise screening questionnaires | ✓ | | ✓ | ✓ | ✓ | ✓ |
| Food intake assessed by a 7 day food diary | ✓ | | ✓ | ✓ | ✓ | ✓ |
| 24-hour blood pressure and capillary or continuous glucose monitoring | ✓ | | ✓ | | | |
| Urine‡ and stool sample | ✓ | ✓ | ✓ | | | ✓ |
| Fat biopsy | | ✓ | ✓ | | | |
| Liver, muscle and gut biopsy | | ✓ | | | | |
| Whole body DEXA | ✓ | | ✓ | | | |
| Sperm sample (male participants only) | ✓ | | ✓ | | | |
| Whole body MRI | ✓ | | ✓ | | | |
| Endoscopy and colonoscopy + biopsy (selected participants only) | ✓ | | ✓ | | | |

*Anthropometrics includes height, waist, hip and neck circumference.
†Blood tests include glucose, insulin, C-peptide, gut hormones, free fatty acids, bile acids, plasma metabolomics, DNA (Visit 1 only), RNA, cortisol, HbA1c, lipids and FGF 19/21.
‡Urine testing will include microalbumin:creatinine and metobolomic assessment.
DEXA, dual-energy X-ray absorptiometry; DNA, deoxyribonucleic acid
; FGF, fibroblast growth factor; HbA1c, glycated haemoglobin; MRI, magnetic resonance imaging; RNA, ribonucleic acid.

► Histological scoring of specimens for non-alcoholic fatty liver disease (NAFLD)/non-alcoholic steatohepatitis (NASH).
► Changes in gut microbiotal diversity, relative abundance of selected phyla (eg, Bacteroides and Firmicutes), and frequency of operational taxonomic units.

### Sample size calculation

Using a logistic regression model to predict a 75% partial and completed diabetes remission rate from multiple predictors, a sample size of 150 provides 90% power at the two-sided 5% significance level to detect an OR of 2.0 per SD of any continuous predictor.[52] This allows for up to 10% of the variation in any one predictor to be explained by the others in the model, and 15% loss to follow-up. The area under the receiver operating characteristic (ROC) curve, overall performance measure, will be estimated with a 95% CI width of at most±0.1 for an area above 0.65. With data analysed from 100 patients there would still be in excess of 80% power to detect the OR of 2.0, and a precision of at most±0.12 for the ROC curve area. This is reasonable enough to allow some deviation in assumptions, such as the dropout rate.

### Measures

Measures are summarised in table 2.

In addition to a full clinical assessment the following procedures will be undertaken:

### Mixed meal test

Preoperatively and at 15 months following their first study visit participants will undergo a mixed meal test, using a stimulus of 14 g protein, 12.9 g fat, 39.6 g carbohydrate, 330 kcal, 137.5 mL (Ensure Compact, Abbott). Prior to this, withdrawal of diabetes medications which will occur using a standardised pathway. Participants will have blood samples taken to measure HbA1c, fasting and stimulated insulin/C-peptide/glucose, gut hormones, free fatty acids, bile acids, plasma metabolites and FGF (fibroblast growth factor) 19/21.

### Questionnaires

The following questionnaires will be used to examine psychological well-being: Alcohol Use Disorders Identification Test (AUDIT), Eating Disorder Examination Questionnaire (EDEQ), Difficulty in Emotion Regulation Scale (DERS), Hospital Anxiety and Depression Scale (HADS) and Short Form 36 (SF-36). The Temperament

**Table 3** Candidate predictors of diabetes remission. V1 is study visit 1

| Predictor | Units/categories | Variable type | Source data | Previous studies |
|---|---|---|---|---|
| Age | years | Continuous | Clinical history | 46 47 56 57 |
| BMI or weight | kg/m$^2$ or kg | Continuous | V1 | 58 59 |
| C-peptide | pmol/L | Continuous | Mixed meal test | 46 56 60 |
| Diabetes duration | years | Continuous | Clinical history | 46 56 57 61–65 |
| Fasting plasma glucose | mmol/L | Continuous | Blood test | 66 |
| Fat distribution | Visceral fat and android: gynoid fat | Continuous | DEXA, impedance, MRI | |
| Genetic risk score | Weighted genetic risk score | Categorical | DNA V1 | |
| HbA1c with medications | mmol/mol<br><br>Insulin, sulphonylureas, insulin sensitising agents, GLP-1 analogues, DPP-IV inhibitors, SGLT-2 inhibitors | Continuous: will interact with medications since directly affected by them | Blood test | 47 57 58 62–65 67–69 |
| HOMA 2 | Using interactive 24-variable assessment of homoeostatic model using default settings | | Mixed meal test | 70 |
| Hypertension | Present/absent<br><br>Controlled by 1–5 agents | Categorical | Medical history, 24-hour blood pressure monitoring | |
| Insulinogenic index | δinsulin (0–30 min) / δglucose (0–30 min) | Continuous | Mixed meal test | |
| Presence of fatty liver disease as assessed by non-invasive markers | NAFLD liver fat score, fatty liver index, FIB-4 | Categorical | Medical history, MRI, non-invasive markers including liver function tests | 71 |
| Presence of diabetes complications | Microvascular disease (including subgroups)<br><br>Macrovascular disease (including subgroups) | Categorical | Clinical history, microalbumin: creatinine, retinal imaging and clinical examination. | 56 |
| Presence of ectopic fat | % liver fat<br><br>% muscle fat<br><br>% pancreatic fat | Continuous | MRI | 72 |
| Sex | Male/female | Categorical | Clinical history | |
| TCI-R scoring | Novelty seeking, harm avoidance, reward dependence, persistence, self-directedness, cooperativeness and self-transcendence | Continuous | V1 | |
| Waist:hip ratio | Measured in cm | Continuous | V1 | |

DPP-IV, Dipeptidyl peptidase-IV ; FIB-4, Fibrosis-4 score; GLP-1, Glucagon-like peptide 1; HOMA, Homeostatic model assessment ; MRI, magnetic resonance imaging; NAFLD, Non-alcoholic fatty liver disease; SGLT-2, sodium-glucose co-transporter-2; TCI-R, Temperament and Character Inventory-Revised.

and Character Inventory-Revised (TCI-R) will also be collected at baseline alone. The International Physical Activity Questionnaire (IPAQ) short form will be used to assess exercise levels.

Questionnaires will be analysed to assess if they impact on likelihood of obtaining remission and, in subsequent visits if they modify the consequences of surgery, that is,

maintenance of diabetes remission. The TCI will be used to assess whether personality subtype predicts metabolic response to surgery.

### Metabolic status

Lipid profile along with 24-hour blood pressure monitoring will be analysed to further assess the response of

these cardiometabolic risk factors to surgery. Continuous glucose measurement sensors or capillary glucose readings will be undertaken to give an alternative measure of glycaemic control.

## Body composition
All patients will undergo a whole body DEXA prior to and 15 months following surgery. This is a validated tool to estimate visceral adipose tissue.[53 54]

Where feasible, participants will undergo a gold standard assessment of visceral adipose tissue using MRI which will also quantify liver, muscle and pancreatic fat.

## DNA
Genomic DNA will be extracted from whole blood samples collected in an EDTA tube. A microarray customised to variants known to be associated with obesity, T2DM and fat patterning will be used to analyse samples. A weighted genetic risk score, with each variant weighted according to their previously published effect size,will be calculated individually.

## Omics
These will be used to explore possible mechanisms explaining the metabolic changes before and after surgery.

### Transcriptomics/epigenomics/proteomics
Longitudinal tissue samples will be collected where possible and will undergo multi-omics assessment.

### Microbiotal analysis
Longitudinal samples of stool will be collected to assess the microbiome.

### Metabolomics
Longitudinal fasting urine and plasma metabolomic samples will be collected pre and following surgery.

## Predictors
Candidate predictors (for examples, see table 3) have been selected based on those identified in previous retrospective cohort studies. Two unique predictors, fat distribution and genetic risk score, will also be tested. We will then cautiously extend the model to address other factors in a univariate manner.

A logistic regression model predicting diabetes remission assessed at the 15 months visit will be built using the prospective cohort data. This was chosen in preference to an ordinal regression approach, such as the proportional odds model, because of the need to have a clear and independent interpretation of the predictive contribution both to complete remission and to complete and partial remission. Logistic regression was preferred over another alternative approach, discriminant analysis, for the same reason and because logistic regression provides predicted probabilities for individual patients which can then be used within ROC curve analysis to provide estimates of sensitivity and specificity of the regression

models measuring their accuracy to predict the diabetes remission outcome of metabolic surgery.

Initially, the effect of each of the clinical predictors and other covariates (patient demographic factors) of interest on each outcome is to be fitted in univariable models. Any clinical predictor that is significant at a threshold of p value≤0.20 will then be included in a multivariable model predicting the outcome of diabetes remission. Clinical predictors and other covariates will then be eliminated in a stepwise fashion until each remaining predictor is significant at the p value=0.05 threshold and the models are found to be stable. Pattern of fat storage will be measured by visceral adipose tissue (VAT) mass, SAT mass and VAT/SAT ratio, and genes influencing diabetes risk and fat deposition will be measured by the Genetic Risk Score using the approach of Fava et al.[51] These will be added to the model, if not already included, to allow their association with diabetes remission to be formally quantified and tested. We will cautiously extend the model to address other factors in a univariate manner. Internal validation of the selection of the predictive model will be done using bootstrap sampling with re-application of the model selection procedure using at least 1000 replications. The performance of the prediction model under different scenarios will be assessed, primarily by obtaining fitted probabilities of remission from the model to estimate the area under the ROC curve, and also by demonstrating the sensitivity, specificity, positive predictive value and negative predictive value across a series of cut points representing a range of scenarios including high sensitivity, high specificity and equivalent sensitivity and specificity.

Survival curves will be plotted to show the proportion remaining free from diabetes over the longer-term according to categories of predictors identified to be important.

### Patient and public involvement
Patients were consulted about the study objectives and selected study procedures to assess acceptability.

## ETHICS AND DISSEMINATION
This study complies with the Declaration of Helsinki and the principles of Good Clinical Practice. The researchers plan to disseminate results at conferences, in peer-reviewed journals as well as lay media and to patient organisations.

**Contributors** JSK, TT, SB, AB, AA, SP and ATP developed the protocol. JSK drafted the paper and TT, SB, AB, ARA, SP, KM and ATP provided critical revisions.

**Funding** This work was supported by NIHR grant number (DRF-2017-10-042). JSK is funded by a National Institute for Health Research (NIHR) Doctoral Research Fellowship for this research project. The views expressed are those of the author(s) and not necessarily those of the NIHR or the Department of Health and Social Care.

**Competing interests** None declared.

**Patient and public involvement** Patients and/or the public were involved in the design, or conduct, or reporting, or dissemination plans of this research. Refer to the Methods section for further details.

**Patient consent for publication** Not required.

**Provenance and peer review** Not commissioned; externally peer reviewed.

**ORCID iDs**
Julia S Kenkre http://orcid.org/0000-0001-7505-4319
Tricia Tan http://orcid.org/0000-0001-5873-3432

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
