## [Reviewer comments · BMJ Open]

ARTICLE DETAILS

TITLE (PROVISIONAL)	Who will benefit from bariatric surgery for diabetes? A protocol for an observational cohort study
AUTHORS	Kenkre, Julia; Ahmed, Ahmed; purkayastha, sanjay; Malallah, Khalefah; Bloom, Stephen; Blakemore, Alexandra; Prevost, A; Tan, Tricia

VERSION 1 – REVIEW

REVIEWER	Miguel A. Rubio Hospital Clínico San Carlos Madrid (Spain)
REVIEW RETURNED	18-Aug-2020

GENERAL COMMENTS	There are several prospective studies that have focused on predictors of diabetes remission. Therefore, this protocol adds other elements to those already known (genetics, omics, ectopic fat ...). Some issues are not sufficiently explained or justified: 1.- Why do not include BMI 30-35 kg/m²? Metabolic surgery also encompasses this BMI range. In fact, most of patients with type 2 diabetes mellitus are found in this BMI range. Certainly, genetic or omic modifications are more interesting in lower than in other higher BMI ranges..2.- It is necessary to specify the RYGB surgical technique. The length of the biliopancreatic and alimentary loops are essential in the reduction and maintenance of body weight as well as in the remission of diabetes. Will the same type of RYGB will be performed to subjects with BMI 35-40 kg/m² as BMI > 50 kg/m² ? The same question arise for insulin-dependent patients, with a long diabetes duration versus patients with non-insulin-dependent treatment.3.- The predetermination of the sample size does not take into account a stratification by BMI, type of treatment (insulin or not), age or gender. These items would consider a limitation of the study that entail a selection bias..4.- 7.5% loss to long-term follow-up is much lower than that described in the literature. 15-20% would be more realistic. How long are the recruitment period?5.- Genetic determinations are poorly described.
--

REVIEWER	Rogério Friedman Endocrine Unit Hospital de Clinicas de Porto Alegre Universidade Federal do Rio Grande do Sul Brazil
REVIEW RETURNED	01-Sep-2020

GENERAL COMMENTS	Thank you for the opportunity to review this interesting manuscript. A
--

	prospective study of the metabolic impact of bariatric surgery, namely, diabetes remission and relapse, has been an unfulfilled need to this date. So, such a project is welcomed. With the intent to contribute to the project, may I offer a few comments. 1) The inclusion criteria mention "stable weight for at least 3 months"; by "stable", do you actually mean "unchanging"? I would rather establish a numerical parameter, for example, "no more than XXX % variation in body weight for at least 3 months". 2) Please inform what bariatric surgery modalities will be included in the study. RYGB has a different impact on the GI tract than does Sleeve gastrectomy. If more than one technique is admitted, please bear into account that you will have to control results for the type of surgical intervention. 3) Different centers often report different results. It would be interesting to have the clinic where surgery was performed as another possible variable to be taken into account (besides the surgical technique). 4) Among the predictors, you correctly include the presence of diabetes complications, both micro- and macrovascular. You mention clinical history as the source of these data, and quote the study of Ugale et al. The study by Ugale and collaborators was retrospective, and based on medical records. Medical records, in many health systems, may lack data on long-term complications of diabetes. Now, your study is prospective. It would be worthwhile to ascertain the presence or absence of diabetes complications by recognized methods (fundus examination or photographs, neurological, albuminuria and so on) just before surgery, and perhaps along the follow up. 5) A logistic regression approach is adequate for the proposal. My concern is the moment of assessment of the primary outcome. Based on retrospective studies, the largest proportion of diabetes remission would happen between twelve and fifteen months, But additional patients will remit only after 18, 24 and even 30 months. It would be most interesting to add 1 or 2 extra time points for verification of the primary outcome, and, afterwards, take into account the length of time required for remission as another variable. 6) Should you take my comments into consideration, please check to what extent the required sample size would be impacted upon. Wishing you success with your proposal.
--	--

VERSION 1 – AUTHOR RESPONSE

Reviewer 1 Comments	
Why do not include BMI 30-35 kg/m2? Metabolic surgery also encompasses this BMI range. In fact, most of patients with type 2 diabetes mellitus are found in this	We completely agree that the lower BMI range is an important one to investigate with respect to metabolic surgery. However, we are limited by the surgery funded on the

BMI range. Certainly, genetic or omic modifications are more interesting in lower than in other higher BMI ranges..	National Health Service in the UK currently. This does include lower BMIs, but only in the South Asian population or in those with recent onset diabetes. Therefore, whilst we expect this will be a minority of our cohort, these patients will be included within the study. See page 10 inclusion criteria
It is necessary to specify the RYGB surgical technique. The length of the biliopancreatic and alimentary loops are essential in the reduction and maintenance of body weight as well as in the remission of diabetes. Will the same type of RYGB will be performed to subjects with BMI 35-40 kg/m2 as BMI> 50 kg/m2 ? The same question arise for insulin-dependent patients, with a long diabetes duration versus patients with non-insulin-dependent treatment.	Thank you. A standard surgical procedure will be performed with a Biliopancreatic limb length of 50 cm and Alimentary limb length of 100cm. We would not differ the procedure for BMIs higher than >50 kg/m². However, it is likely that those with very high BMIs may be offered RYGB as part of a two stage procedure – in those cases we would not be able to recruit patients in this population. Please see modification on page 10 Regarding the second part of your question, no variation in the limb lengths would be made based on the diabetes severity or duration. This is the current practice within the units planned to participate in this study.
The predetermination of the sample size does not take into account a stratification by BMI, type of treatment (insulin or not), age or gender. These items would consider a limitation of the study that entail a selection bias..	The sample is a consecutive sample and will include all eligible patients at the Imperial Weight Centre or Chelsea and Westminster NHS Trust. Therefore we believe the sample will be representative of its population, and there is not a need to change the sample size determination. A resulting model with novel covariates would be researched further in a larger population, which may differ according to these factors. Please see modification on page 9
7.5% loss to long-tern follow-up is much lower than that described in the literature. 15-20% would be more realistic. How long are the recruitment period?	Thank you, as you rightly point out it is difficult to estimate the likely loss-to-follow up rate. We have amended the sample size calculation to allow for an estimated 15% drop-out rate, and have indicated that a higher dropout rate would still be consistent with in excess of 80% power and reasonably precision for estimating the model's performance. We have made a revision to the manuscript to update this. Our recruitment period is planned to be up to 3 years. See modification of section entitled 'Sample size' page 11
Genetic determinations are poorly described.	Thank you. We have made revisions to the text to improve the clarity of this section. Please see modified DNA section on page 16
Reviewer 2 Comments	
The inclusion criteria mention "stable weight for at least 3 months"; by "stable", do you actually mean "unchanging"? I	This parameter is assessed at their screening visit for study inclusion. We would assess 'stable' as self-reported stable weight over the

would rather establish a numerical parameter, for example, "no more than XXX % variation in body weight for at least 3 months".	preceding 3 months equating to no more than 10% weight loss. See modification of inclusion criteria page 10
Please inform what bariatric surgery modalities will be included in the study. RYGB has a different impact on the GI tract than does Sleeve gastrectomy. If more than one technique is admitted, please bear into account that you will have to control results for the type of surgical intervention.	We completely agree that weight loss and mechanism of diabetes remission likely varies amongst procedures. Patients will be undergoing RYGB alone. This has been clarified in the manuscript. See clarification on page 10
Different centers often report different results. It would be interesting to have the clinic where surgery was performed as another possible variable to be taken into account (besides the surgical technique).	To limit variation in technique surgery will be standardised and performed by a limited pool of surgeons trained with the same technique. It would certainly be interesting to examine the remission results from surgeons at other centres, but this is outside of the remit of this study.
Among the predictors, you correctly include the presence of diabetes complications, both micro- and macrovascular. You mention clinical history as the source of these data, and quote the study of Ugale et al. The study by Ugale and collaborators was retrospective, and based on medical records. Medical records, in many health systems, may lack data on long-term complications of diabetes. Now, your study is prospective. It would be worthwhile to ascertain the presence or absence of diabetes complications by recognized methods (fundus examination or photographs, neurological, albuminuria and so on) just before surgery, and perhaps along the follow up.	This is a very important point. Our clinical records include retinal imaging and we plan to obtain this for all participants. We also are collecting microalbumin: creatinine. We have updated the manuscript to update this. Please see modification on page 14 -15, Table 2
A logistic regression approach is adequate for the proposal. My concern is the moment of assessment of the primary outcome. Based on retrospective studies, the largest proportion of diabetes remission would happen between twelve and fifteen months, But additional patients will remit only after 18, 24 and even 30 months. It would be most interesting to add 1 or 2 extra time points for verification of the primary outcome, and, afterwards, take into account the length of time required for remission as another variable.	Thank you, to ensure deliverability of the study we have chose 15 months as our endpoint. However, we are collecting fasting glucose, insulin, and HbA1c data at 2, 3 and 5 years. We note at the very end of the document that 'Survival curves will be plotted to show the proportion remaining free from diabetes over the longer term according to categories of predictors identified to be important.' Please see page 21
Should you take my comments into consideration, please check to what extent the required sample size would be impacted upon.	Thank you for your comments. The sample size is primarily determined by the number of covariates included in the statistical model. This will necessarily depend on any covariates of interest identified during the initial univariable analysis.

VERSION 2 – REVIEW

REVIEWER	Rogério Friedman Federal University of Rio Grande do Sul (Hospital de Clínicas de Porto Alegre). Brazil.
REVIEW RETURNED	13-Dec-2020
GENERAL COMMENTS	Thank you for taking into consideration my suggestions and remarks. The revised manuscript has resulted into an improved project proposal.